# Updates on the Functions and Molecular Mechanisms of the Genes Involved in *Aspergillus flavus* Development and Biosynthesis of Aflatoxins

**DOI:** 10.3390/jof7080666

**Published:** 2021-08-17

**Authors:** Elisabeth Tumukunde, Rui Xie, Shihua Wang

**Affiliations:** Key Laboratory of Pathogenic Fungi and Mycotoxins of Fujian Province, Key Laboratory of Biopesticide and Chemical Biology of Education Ministry, and School of Life Science, Fujian Agriculture and Forestry University, Fuzhou 350002, China; 1161917003@fafu.edu.cn (E.T.); 2180514015@fafu.edu.cn (R.X.)

**Keywords:** *A. flavus*, aflatoxins, gene, secondary metabolites, biosynthesis

## Abstract

*Aspergillus flavus* (*A. flavus*) is a ubiquitous and opportunistic fungal pathogen that causes invasive and non-invasive aspergillosis in humans and animals. This fungus is also capable of infecting a large number of agriculture crops (e.g., peanuts, maze, cotton seeds, rice, etc.), causing economic losses and posing serious food-safety concerns when these crops are contaminated with aflatoxins, the most potent naturally occurring carcinogens. In particular, *A. flavus* and aflatoxins are intensely studied, and they continue to receive considerable attention due to their detrimental effects on humans, animals, and crops. Although several studies have been published focusing on the biosynthesis of the aforementioned secondary metabolites, some of the molecular mechanisms (e.g., posttranslational modifications, transcription factors, transcriptome, proteomics, metabolomics and transcriptome, etc.) involved in the fungal development and aflatoxin biosynthesis in *A. flavus* are still not fully understood. In this study, a review of the recently published studies on the function of the genes and the molecular mechanisms involved in development of *A. flavus* and the production of its secondary metabolites is presented. It is hoped that the information provided in this review will help readers to develop effective strategies to reduce *A. flavus* infection and aflatoxin production.

## 1. Introduction

*Aspergillus flavus* (*A. flavus*) is a common saprophytic fungus. Classically, *A. flavus* belongs to the kingdom of Fungi, phyllum of Ascomycota, and order of Eurotiales in the Eurotiomycetes class. It also belongs to the family of Trichocomaceae and the genus *Aspergillus* in species *flavus* [1]. This fungus is known to be an opportunistic pathogen, and it is capable of infecting a number of seed crops (e.g., peanuts, maze, cotton seeds, rice, etc.) and living organisms including animals and humans [2,3]. In general, *A. flavus* produces highly toxic secondary metabolites named aflatoxins, and the consumption of highly aflatoxin contaminated food can cause death or chronic diseases particularly in immune-suppressed patients. The discovery of aflatoxins in 1961 gave rise to modern mycotoxicology and stirred up scientific interest [4]. Studies on aflatoxins led to a golden age of mycotoxin research during which numerous other new mycotoxins, such as fumigillin, aspertoxin, cyclopiazonic acid, aflatrem, ochratoxin, and patulin, were also discovered [5,6]. Among all polyketides and mycotoxins produced by fungal species, aflatoxins are known as the major mycotoxins, including aflatoxin B1 (AFB_1_), B*2* (AFB_2_), G1 (AFG_1_), and G2 (AFG_2_). Additional aflatoxins, such as M1 (AFM_1_) and M2 (AFM_2_), which are the metabolites from AFB_1_ and AFB_2,_ respectively, are generally found in cow milk in areas of high aflatoxin exposure. Subsequently, humans may be exposed to these aflatoxins through milk and milk products, including breast milk, especially in areas where the lowest quality grain is used for animal feed. Of all the aflatoxins identified so far, AFB_1_ has been found to the most toxic and foremost potent carcinogen natural chemical compound produced by *Aspergillus* fungi (e.g., *A. flavus*, *A. nomius*, *A. parasiticus*, etc.) [2,7].

Aflatoxin biosynthesis is a process that involves multiple enzyme reactions with several bioconversion steps. In recent years, worldwide efforts were made to better understand the biochemistry, genetics, and regulation of aflatoxin biosynthesis [8,9]. In the last decade, for example, significant progress has been made in discovering the genes and their enzymes that are involved in each step of the aflatoxin biosynthetic pathway [10,11], further confirming that the aflatoxin biosynthesis pathway is one of the most studied fungal secondary metabolite pathways in *A. flavus* [12,13]. Although the aflatoxin biosynthesis pathway in *A. flavus* has been extensively studied, its molecular mechanisms are still not well known. As a result, this review focuses on the recently discovered functions and molecular mechanisms of genes involved in fungal development (e.g., production of conidiophores, conidia, sclerotia, mycelia growth, etc.) and biosynthesis of aflatoxins in *A. flavus*. In doing so, updated information about *A. flavus* genomics and its conserved transcription factors involved in morphogenesis and aflatoxins production is provided. Then, an overview on posttranslational modifications (e.g., methylation, acetylation, phosphorylation, sumoylation, succinylation, etc.), signal pathways (e.g., HOG, Slt2, etc.), transcriptome, and metabolomics involved in morphogenesis and aflatoxin production in *A. flavus* is also provided. To keep our discussion concise, a special focus is directed to the studies that have been published in the last 10 years.

## 2. *A. flavus* Genomics and Aflatoxins Biosynthesis Genes

The term genomics denotes the process of revealing the whole genetic content of any kind of organism through the bioinformatics analysis of sequences of the genomic DNA. Additionally, the fungal genomic information allow scientist to analyze the revolutionary relationship between different species [14]. Interestingly, *A. flavus* genome is already known, and its whole genome microarray is also available [15]. The availability of the genome sequence of *A. flavus* has enabled significant efforts in the understanding of the regulation of aflatoxin biosynthesis and the establishment of new control strategies [16]. Indeed, the availability of *A. flavus* genomic data marked the beginning of a new research phase not only in fungal biology but also in other fields, such as medical mycology, pathogenicity, agricultural ecology, aflatoxins biosynthesis, and evolution. The genome microarrays can be used to identify genes responsible for the different mycotoxins’ (typically aflatoxins) biosynthesis and fungal infections in living organisms [17,18]. Several studies [19,20] indicated that genome of *A. flavus* NRRL 3357 consists of eight chromosomes and has an estimated size of approximately 36.8 Mega base pairs (Mb) with about 12,197 predicted number of genes. Recently, a chromosome-level assembly of the *A. flavus* genome was completed [21]. In 2006, Bhatnagar et al. [22] reported that genome of *A. flavus* is compressed with several duplicates of genes or less copied sequence. In fact, *A. flavus* consists of at least 56 secondary metabolic gene clusters containing about 12,000 predicted functional genes that could regulate the production of abundant secondary metabolites, including aflatoxins [18].

Until now, about 30 genes have already been identified as being integral components of aflatoxin biosynthetic gene cluster in *A. flavus*, and all of them are clustered in a 70 kb DNA region on the chromosome 3, named the toxin-producing gene cluster [7,23]. All these genes contain the same regulatory trend, and their expression levels can directly affect the level of aflatoxin biosynthesis [24]. The aflatoxin pathway genes are mainly symbolized by a code of three-letter “afl” followed by a capital letter from “*A*” to “Y” characterizing each individual gene potentially involved or confirmed to be involved in the aflatoxin biosynthesis process. Therefore, different aflatoxin biosynthesis pathway genes were named starting from *aflA* to *aflY*. The aflatoxin biosynthesis pathway also involves sterigmatocystin (ST) synthesis genes, including *a**flA* to *aflQ*, which start from the first conversion of fatty acid to the final product, and some other genes whose involvement in this pathway is still unclear to this date [12,18]. In particular, AFB_1_ has been found to be originated from a polyketide precursor, and it has been generally established to be synthesized following a path that was previously described by Caceres et al. [24]. However, it has been suggested that other aflatoxins, including AFB_2_, AFG_2_, and AFG_1_, could be metabolically linked by a direct interconversion method [18,25]. To this end, many enzymes do actively contribute to the conversion of the precursors at different stages involved in the aflatoxin biosynthesis pathway [25].

Consistent with the aforementioned, aflatoxins biosynthesis requires an intricate regulatory mechanism that is composed of several specific regulatory genes in the pathway. In particular, both *aflR* and *aflS* have been reported to be major aflatoxin transcriptional regulator genes [15,26]. A number of studies conducted by our research group on the genes functions [27,28,29,30,31] indicated that several other structure genes are equally related to the aflatoxin production in *A. flavus*. For example, after the deletion of an antioxidant catalase *cta1*, the expression levels of *aflQ*, *aflC*, and *aflD* were down-regulated, showing that *cta1* plays an important role in aflatoxin biosynthesis in *A. flavus* [27]. The nucleoside diphosphate kinase (*Aflndk*) was also found to play an essential role in sclerotia production and conidia development in *A. flavus* [28]. Specifically, all *Aflndk* mutant strains displayed visible effects on sporulation, including the atrophy of spore heads and stalks, while the amount of conidia and sclerotia production decreased. In [29], we found that the deletion of *hexA* gene significantly reduced the production of AFB_1_, conidia, and conidiophores, while the deletion of *sakA* significantly decreased the mycelia growth and increased both sclerotia and AFB_1_ production [13]. In [30], we also found that the deletion of *pex5* caused defects in sclerotial formation, sporulation, stress response, carbon metabolism, crop infection, and aflatoxin biosynthesis.

## 3. *A. flavus* Transcription Factors Involved in Morphogenesis and Aflatoxin Production

Transcription factors (TFs) are known as sequence-specific DNA-binding factors controlling transcription rate of genetic information from DNA to mRNA through binding to specific DNA sequences [32]. A recent study [33] found that TFs are the key regulatory proteins in living organisms. In fungal species, TFs include either proteins that are encoded by cluster-associated (pathway-specific) regulatory genes or broad-domain regulatory proteins, which can affect the expression of several genes involving in biosynthetic procedures [33]. In this case, TFs are considered the final connection between the target genes and the signal flow that play important roles in the signal transduction pathways. On the basis of conserved DNA-binding domain, TFs are normally classified into different categories and structures [33,34]. Up to now, several studies have identified the roles of different TFs in *A. flavus*. Cary et al. [35] reported the function of the Homeobox (*hbx1*) transcriptional factors in *A. flavus*. They found that disruption of *hbx1* gene leads to loss of production of sclerotia, conidiophores, and conidia as well as aflatrem, cyclopiazonic acid, AFB_1_, and AFB_2_. They further observed significant down-regulation in the expression levels of three aflatoxins biosynthetic genes (*aflM*, *aflD*, *aflC*) and one regulatory gene (*aflR*) in *A. flavus*. A latter study [36] confirmed that *hbx1* gene can affect the expression of several genes in the aflatoxins biosynthesis pathway and the expression levels of transcription factor genes involved in *A. flavus* development, including conidiophores biogenesis genes (*flbA*, *flbC*, *flbD*, and *flbE*) and conidiation regulatory pathway genes (*brlA* and *wetA*). A research conducted by Hu et al. [37] found the plant homeodomain (PHD) transcription factor *Rum1* to be a regulator of morphogenesis and aflatoxin biosynthesis in *A. flavus*. Another study discovered that basic leucine zipper (bZIP) transcription factor *AflRsmA* could regulates sclerotium formation, oxidative stress response, and AFB_1_ biosynthesis in *A. flavus* [34]. It has also been revealed that the *Skn7* transcription factor influences aflatoxin production, pathogenicity, morphological development, and stress response in *A. flavus* [38]. In addition, in *A. flavus*, APSES transcription factors *AfStuA* and *AfRafA* were shown to play important roles in pathogenicity, fungal development, and mycotoxin synthesis [39]. Cary et al. [40] examined the function of transcription factors *NsdD* and *NsdC* on the development and aflatoxin production in *A. flavus*, and it was found that *NsdD* and *NsdC* affect the transcription of genes that are essential for sclerotial morphogenesis and aflatoxin biosynthesis in *A. flavus*. These TFs play important roles in morphogenesis and aflatoxin production in *A. flavus*.

## 4. Aflatoxins Production at the Transcriptome Level in *A. flavus*

The term transcriptome is derived from the two words transcript and genome, and it is linked with the process of transcript formation during transcription. Two biological techniques are generally used in the study of transcriptome, including RNA sequencing (RNA-seq) (i.e., sequence-based approach) and DNA microarray (i.e., a hybridization-based technique). However, compared to the outdated DNA microarray, RNA-seq is the most preferred and dominant transcriptomics method because it generally detects higher percentages of differentially expressed genes, especially the genes with low expression (i.e., can quantify expression across a larger dynamic range (>10^5^ for RNA-Seq vs. 10^3^ for DNA microarray)) [41,42,43]. A list of the key advantages of RNA-Seq compared with DNA microarray method is provided in [41], and interested readers are directed to this specific study for more information.

Consistent with the aforementioned, several studies were conducted to understand the effect of environmental conditions on the development of *A. flavus* and its aflatoxin production at the transcriptome level. In a study conducted by Zhang et al. [44], the RNA-seq approach was used to delineate *A. flavus* transcriptome and specific data about differentially expressed genes under different water activities (0.99 and 0.93 a_w_). It was found that 0.93 a_w_ significantly decreased conidiation and aflatoxins biosynthesis in *A. flavus* compared to 0.99 a_w_. Among the 33 genes identified in the transcriptome of *A. flavus*, the expression of 16 genes related to the production of aflatoxins were up-regulated more than twofold under 0.99 a_w_ compared to 0.93 a_w_. In the same study, the expression of 11 genes related to the development of *A. flavus* was increased after 0.99 a_w_ treatment, suggesting that *A. flavus* undergoes a substantial transcriptome response during a_w_ variation.

In Yu et al. [45], the authors applied the RNA-seq approach to profile the *A. flavus* transcriptome under various temperature conditions and identified a number of differentially expressed genes under aflatoxin conducive and non-conducive conditions. More specifically, some of these identified genes (e.g., *aflT*, *aflC*, *aflNa*, *aflx*, *sugR*, *glcA*, *hxtA*, *nadA*, etc.) were found to be involved in aflatoxin biosynthesis and adjacent sugar utilization, while others (e.g., *maoA*, *dmaT*, *pks-nrps*, etc.) were involved in cyclopiazonic acid transportation and aflatoxin biosynthesis. In a similar study, the RNA-seq method was also used to explore the transcriptome of *A. flavus* in response to the treatment with 5-Azacytidine (5-AC) [46]. 5-AC is a derivative of the nucleoside cytidine that is used in epigenetics as an inactivator of DNA methyltransferase. When *A. flavus* was treated with 5-AC, 240 genes were significantly expressed, and these included the genes that are involved in *A. flavus* development (e.g., *veA*, *flbA*, s*tuA*, *abaA*, *brlA*, *rodA*, *wetA*, *nsdD*, *rodB*, etc.) and aflatoxin biosynthesis (e.g., *aflQ*, *aflX*, *aflI*, *aflLa*, *aflG*, etc.). They also include the genes that are commonly involved in proteolytic functions and catalytic activities (e.g., AFLA_000860, AFLA_004460, AFLA_008370, AFLA_011740, AFLA_121360, etc.). The latter regulate the modification of some essential proteins participating in the formation of aflatoxins in aflatoxisomes (i.e., vesicles that generate aflatoxins) [46]. A study conducted by Yao et al. [47] examined the regulatory mechanism of aflatoxins using comparative transcriptomics in *A. flavus* and identified 18 novel (i.e., AFLA_084720, NosA, AFLA_019420, AFLA_081920, AFLA_108250, AFLA_108260, AFLA_131810, AFLA_019430, AFLA_138920, AFLA_132980, AFLA_039650, AFLA_108810, AFLA_065960, AFLA_013540, and four others genes whose identities were not provided) and 12 known (i.e, *meaB*, *rtfA*, *atfB*, *gprP*, *pdeH*, *pdeL*, *laeB*, *ppoC*, *hamE*, *hamF*, *hamG*, and *hamH)* regulatory genes that are essential for aflatoxin biosynthesis. In their study, the authors found that the deletion of an *LaeA*-like methyltransferase (*Lael1*) from *A. flavus* resulted in a significant increase in AFB_1_ production. They also found that the inactivation of both *AfStuA* and *AfRafA* in *A. flavus* significantly decreased the level of AFB_1_ production. Bai et al. [48] showed 135 differentially expressed miRNA-like RNAs under different temperature and a_w_.

In summary, transcriptomics helps in the determination of the transcription of the genes and the quantification of changes in the expression levels of every transcript during the cell development and aflatoxin production under different environmental conditions. To this end, it is important to determine the transcriptome in molecular biology (i.e., it captures a snapshot in time of the total transcripts present in a cell) to guarantee an accurate determination of the cells’ molecular constituents and, subsequently, the interpretation of the functional elements of the genome in *A. flavus*.

## 5. Proteomics Analysis

Proteomic analysis (proteomics) is a powerful tool that helps in the identification of the entire complement of proteins (the proteome) and provides a systematic evaluation of protein expression in a biological organism under a particular condition [49]. A previous study also reported proteomic analysis as the most influential method to identify proteins in complex mixtures, and it is applied in the study of changes in protein expression in the organism under different environmental states [50]. Therefore, the development of quantitative, highly accurate, and sensitive proteomic techniques has helped in the discovery of novel proteins and the quantification of their expression. For example, by using isobaric tags for relative and absolute quantitation (iTRAQ) technique, multiple samples of proteins can be analyzed simultaneously [51,52].

Several studies have been conducted by our research group to investigate the responses of *A. flavus* proteome under different environmental conditions [53,54]. In Zhang et al. [53], the authors used the iTRAQ technique to analyze the proteomic alterations in *A. flavus* under two different water activity (a_w)_ levels (0.93 and 0.99 a_w_)). They identified a total of 3566 proteins, of which 837 proteins were differentially expressed in response to the aforementioned a_w_ levels. They found that two of these proteins, namely AFL2G_04330 and KapK, play a critical role in the production of aflatoxin. They indicated that AFL2G_04330 protein plays an essential role in the activation of asexual sporulation and the production of aflatoxins in *A. flavus*, while KapK down-regulates aflatoxin biosynthesis. Another study by Bai et al. [54] investigated changes in translation and relative protein levels in *A. flavus* in response to different temperature levels (28 °C and 37 °C). They used the iTRAQ method to identify and quantify the proteins. A total of 3886 proteins were identified, of which 2832 proteins were quantified. A further analysis revealed that 12 of these fully quantified proteins (namely aflE, aflW, aflC, aflD, aflO, aflP, aflK, aflM, aflY, aflJ, aflS, and aflH) were highly expressed at 28 °C, which resulted in an increase in the aflatoxins biosynthesis. In proteogenomic analysis, Yang et al. [30] used several samples from eight different stress treatments and found that several novel genes were differently expressed in response to these stress conditions. They identified 732 novel protein-coding genes, and some of these conserved proteins (e.g., YdiU domain protein, Sac7, nmrA-like family protein, protein trm-112, Cds1, MpkC, etc.) were found to play important functions in regulating the pathogenicity, survival, growth, stress responses, and aflatoxin biosynthesis (e.g., autophagy-related protein 7, NADPH cytochrome P450, lipase/esterase family protein, CFEM protein domain, O-methylsterigmatocystin oxidoreductase, etc.) in *A. flavus*.

Consistent with the aforementioned, it is clear that proteome analysis could be an essential approach in delineating some of the unknown molecular mechanisms contributing to the development and aflatoxins biosynthesis in *A. flavus*. More importantly, the aforementioned iTRAQ technique can be used to reveal changes in gene-product expression during different biological processes, including the fungal development and the secondary metabolite biosynthesis. Indeed, accurate determination of protein expression would undoubtedly contribute to an improved understanding of processes related to the fungal development and secondary metabolite biosynthesis, which could subsequently help in devising control strategies for their reduction and/or elimination.

## 6. Metabolomics Analysis

In general, metabolomics refers to the scientific identification and quantification of chemical substances involving metabolites, the intermediates, small molecule substrates, and metabolism products of an organism. The term metabolome represents the total number of metabolites within an organism, and these are the final products of cellular processes. The *A. flavus* metabolome is of substantial interest since *A. flavus* is capable of producing the potent carcinogenic and toxic metabolites [55,56]. Apart from aflatoxins and their precursors, *A. flavus* produces other secondary metabolites, some of which may be toxic (e.g., cyclopiazonic acid, aflatrem, etc.) [57], while others are important for some chemical processes (e.g., kojic acid) [58]. Although a detailed discussion about these additional secondary metabolites is beyond the thematic topic of the present paper, some important points are worth mentioning. For example, studies [57,59] suggest that cyclopiazonic acid (PCA) does lead to a variety of human diseases, including muscle necrosis, intestinal hemorrhage, and edema oral lesions, while aflatrem causes neurological disorders. In contrast, kojic acid is sometimes used in the food industry and in some health and cosmetic products as a natural preservative [58]. Additional information regarding these additional secondary metabolites as well as their effects and genes involving in their biosynthesis can be found in the literature [57,60,61,62], and interested readers are directed to these specific studies for more information.

In order to determine and/or quantify metabolites, metabolomics uses different complementary analytical methodologies, such as GC-MS, LC-MS/MS, and/or NMR [63]. In a recent study, Song et al. [64] used GC-MS analysis to investigate the differences in metabolic changes between *A. flavus* A133 and NT strains treated with 5-AC. Among the 1181 identified volatile metabolites, numerous fatty acid-derived volatiles (e.g., Ergosterol, n-Hexadecanoic acid, Normeperidinic acid, 6-Octadecenoic acid, n-Propyl9-octadecenoate, 6,9-hexadecadienoic acid, Trimethylammonioacetate, etc.) were found to act as important intermediates for the biosynthesis of aflatoxins. The above authors also found that these fatty-acid-derived volatiles affect the balance between the production of conidia and the formation of sclerotia in *A. flavus*. To find out the function of the genes related to secondary metabolite biosynthesis, methods of over-expression or deletion of the target gene followed by metabolite profiling are generally used [65]. Recently, the effect of *rtfA* on the *A. flavus* metabolome was examined, whereby metabolomic analysis demonstrated that *A. flavus rtfA* could affect the production of numerous secondary metabolites, such as AFB_1_, leporins, aflatrem, aspirochlorine, aflavinines, and ditryptophenaline, further confirming the *rtfA* functions as a global positive regulator of secondary metabolites biosynthesis in the *Aspergillus* species [66]. Therefore, metabolomics has the potential to find different metabolites influencing biological processes in *A. flavus*.

## 7. Post Translation Modifications Influencing Development and Aflatoxin Biosynthesis in *A. flavus*

Several studies demonstrated the effects of post-translational modifications (PTMs) in different biological processes (e.g., aflatoxins biosynthesis, development, virulence, etc.) in *A. flavus* [67,68,69]. In this section of the paper, a brief discussion on the influence of different PTMs, such as methylation acetylation, phosphorylation, sumoylation, and succinylation, on development and biosynthesis of aflatoxins in *A. flavus* is provided.

In a previous study, Lee et al. [70] indicated that DNA methylation could play an essential function in the development and production of secondary metabolites in *Aspergillus* species. Gowher et al. [71] found the levels of methylated DNA to be low in *A. flavus*. A later study carried out in our laboratory found that an arginine methyltransferase gene *aflrmtA* is involved in the biosynthesis of AFB_1_ and the morphogenesis and pathogenicity of *A. flavus*. Deletion of *aflrmtA* in *A. flavus* led to a decrease in the production of AFB_1_ and sclerotia but increased the production of conidia and conidiophores. The deletion of *aflrmtA* also decreased expression levels of the AFB_1_ biosynthesis regulatory gene *aflR* and AFB_1_ biosynthesis genes *aflK* and *aflC* [72]. In recent studies conducted by our research group [73,74,75], we demonstrated that both the histone and DNA methyltransferases play a significant role in aflatoxin biosynthesis in *A. flavus*. In particular, the deletion of the *dmtA* gene increase sclerotial production but decrease both conidiation and aflatoxin biosynthesis [73]. In separate studies, we investigated the biological role of the histone methyltransferase *Aflset1* [74] and histone methyltransferase *dot1* [75]. It was found that *Aflset1* is involved in the regulation of virulence, fungal morphogenesis, and AFB_1_ biosynthesis in *A. flavus* for the former and the regulation of pathogenicity and aflatoxin attributes for the latter. In *A. flavus*, acetylation has been shown to involve in sclerotia formation, fungal development, pathogenicity, and aflatoxin biosynthesis [76]. Recently, protein lysine acetylation sites in the *A. flavus* proteome were identified by using a combination of both affinity enrichment and LC-MS/MS analysis method. A total of 1383 unique protein lysine acetylation sites in 652 acetylated proteins were found to be involved in different cellular processes, including secondary metabolite synthesis, cell growth, and gene expression [77]. Among the many different PTMs, lysine acetylation controlled by lysine deacetylases and acetyltransferases is a highly evolutionarily and dynamically conserved PTM occurring in eukaryotes and prokaryotes [78,79]. Genomic analysis revealed that genes encoding acetyltransferases are present in the *A. flavus* genome [77]. Lan et al. [68] found that deletion of histone acetyltransferase *AflgcnE* reduced the growth of *A. flavus* and decreased the hydrophobicity of the cell’s surface. In their study, the authors demonstrated that the *AflgcnE* mutant strain does not produce aflatoxins and is unable to generate asexual sporulation and sclerotia. They also found that *AflgcnE* is required for maintain both cell wall integrity and genotoxic stress responses in *A. flavus* [68]. In a similar study, Lan et al. [80] demonstrated that *SinA* (a transcriptional regulator) interacts with *HosA* histone deacetylase to regulate the development and production of AFB_1_ in *A. flavus*.

By using MS-based proteomics, Bai et al. [81] identified a comprehensive phosphorylation database containing 62,272 non-redundant phosphorylation sites within a total number of 11,222 phosphoproteins from eight fungal species, including *A. flavus*. An additional research was conducted on a site-specific and global phosphoproteomic analysis of *A. flavus*, and 598 high-confidence phosphorylation sites were identified in a total of 283 phosphoproteins. These identified phosphoproteins have been found to be involved in different biological processes, including aflatoxins biosynthesis and signal transduction [82]. A study published by Yang et al. [83] indicated that *cdc14* is involved in the regulation of fungal development, stress response, pathogenicity, and aflatoxin biosynthesis in *A. flavus*. They found that *cdc14* reduces biosynthesis of aflatoxins by suppressing the expression levels of *aflR*, *aflS*, *aflD*, *aflC*, *aflK*, and *aflQ* genes in the aflatoxin gene cluster. A high-affinity phosphodiesterase, *pdeH*, has also been shown to play a regulatory function in development and aflatoxin biosynthesis in *A. flavus* [84]. Another study found that the MAPK-related tyrosine phosphatases plays essential functions in the regulation of secondary metabolism as well as the development and pathogenicity in *A. flavus* [85].

In a research conducted by Nie et al. [67], sumoylation was found to be involved in both toxin attributes and virulence of *A. flavus*. They revealed that deletion of *AfsumO* gene leads to a significant decrease in AFB_1_ and AFB_2_ production and impairs the expression of the genes related to aflatoxin biosynthesis, including *aflP*, *aflO*, *aflD*, *aflC*, *aflA*, *aflS*, and *aflR* in *A. flavus*. Although sumoylation has been extensively studied in higher plants, vertebrates, and yeasts [67,86,87], our literature survey revealed that there is a significantly high paucity of similar studies on *A. flavus*.

In their study, Ren et al. [69] conducted a global analysis of lysine succinylome in *A. flavus* using high resolution mass spectrometry method, and they identified a total of 985 succinylation sites in 349 succinylated proteins. These succinylated proteins were found to be involved in diverse biological processes, including aflatoxins biosynthesis and pathogenicity. The above authors also found that lysine succinylation sites on the norsolorinic acid reductase NorA (*AflE*), which is an essential enzyme in aflatoxins biosynthesis pathway, affects the production of both sclerotia and AFB_1_ production in *A. flavus*.

In summary, these modifications are crucial mechanisms that help to identify the functions of different proteins involved in a number of biological processes in *A. flavus*. In fact, all these modifications can regulate the growth, development, and virulence of *A. flavus*. In this case, they play an important role in promoting epigenetics in fungal development and the production of secondary metabolites. They can also be used for clinical prevention and control of *A. flavus*. Apart from the above-discussed PTMs (i.e., succinylation, sumoylation, phosphorylation, acetylation, and methylation), there are currently many other modifications documented in the literature, though to a limited extent. These include but are not limited to farnesylation, glycosylation, ubiquitination, palmitoylation, neddylation, and proteolysis [88,89,90]. All the studies reviewed in this section demonstrate that by studying the functions of these PTMs, one can then identify their effects on the development and aflatoxins biosynthesis in *A. flavus*.

## 8. Signal Pathways Involved in Morphogenesis and Secondary Metabolism in *A. flavus*

### 8.1. HOG Pathway

The high-osmolarity glycerol (HOG) signaling pathway is one of the mitogen activated protein kinases (MAPKs) pathways in eukaryotic cells. This pathway contains a protein kinase known as HogA, which contributes to the regulation of osmotic stress, and it is also able to recognize all steps that involves in the osmoadaptation process [13,91]. Recently, several studies were conducted by our group on the functions of the genes that are involved in HOG pathway, including *Aflste20*, *pbsB*, and *sakA*/*hogA* [13,92,93]. It was found that deletion of *Aflste20* gene resulted in complete loss of sclerotia, reduced growth, decreased AFB_1_ production, as well as decreased expression of key structural genes *aflD*, *aflQ*, and *aflK* and regulatory genes *aflS* and *aflR* in the aflatoxin biosynthesis cluster in *A. flavus* [92]. In a similar study, it was shown that inactivation of pbsB significantly decreased the production of AFB_1_, conidiation, mycelia growth, and sclerotia production in *A. flavus*. The expression levels of *aflR*, *aflQ*, *aflK*, *aflD*, and *aflC* were also significantly down-regulated in *ΔpbsB* mutant, showing that *pbsB* gene increases AFB_1_ biosynthesis by up-regulating the expression levels of structural and regulatory genes in the aflatoxin gene cluster [93]. The membrane mucin *Msb2* was found to be involved in the regulation of *A. flavus* morphological development, pathogenicity, secondary metabolism, and stresses adaptation [94]. A study on the functions of *sakA*/*hogA* also revealed that this gene regulates the morphology development, pathogenicity, stress response, and reduces biosynthesis of aflatoxins by suppressing the expression levels of genes in the aflatoxin gene cluster in *A. flavus* [13]. All these findings highlight the importance of HOG pathway in the physiological processes in *A. flavus.* A speculated HOG signaling pathway in *A. flavus* is shown in Figure 1.

### 8.2. Slt2 Pathway

A recent study by Zhang et al. [95] demonstrated that the MAP kinase Slt2 modulates the fungal development and pathogenicity as well as the aflatoxin biosynthesis and stress response processes in *A. flavus*. These findings reaffirm the importance of Slt2-MAPK pathway in different physiological processes in *A. flavus*. Another study [96] was been conducted on the functions of MAPKKK Bck1, which is the upstream element of Slt2-MAPK pathway, and it was found that deletion of *Bck1* showed more sensitivity to cell wall stress, decreased pathogenicity on peanut seeds, and also caused a significant defect in growth and development in *A. flavus*. It was also observed that the lack of *Bck1* significantly increased the production of AFB_1_, which indicates that *Bck1* decreases AFB_1_ biosynthesis by down-regulating the expression levels of genes in the aflatoxin gene cluster in *A. flavus*. A speculated Slt2-MAPK signaling pathway in *A. flavus* is shown in Figure 2.

### 8.3. Other Pathways

Apart from the previously discussed pathways, there are some other signal pathways that have been identified to play important roles in diverse biological processes in eukaryotes, including filamentous growth (Kss1), ascospore formation, mating pheromone (Fus3), and cAMP/PKA pathways [91,97,98]. In *A. flavus*, additional pathways were been identified and documented in the literature. To this end, different functions of several important components of cAMP/PKA pathway, including *GpaB*, *PdeH*, *PdeL*, *AcyA*, and *Cap*, were recently identified by our research group. Liu et al. [99] found that G protein alpha, which is a subunit *GpaB*, is required for fungal development, pathogenicity, and aflatoxin biosynthesis in *A. flavus*. Yang et al. [98] indicated that *AcyA* regulates the fungal virulence and development as well as the aflatoxin biosynthesis in *A. flavus*. Additionally, cAMP phosphodiesterases, *PdeH* and *PdeL*, have also been reported as being negative regulators of the synthesis of cAMP and the production of AFB_1_ in *A. flavus* [84]. A recent study by Yang et al. [100] also indicated that Cap containing multi-domain is engaged in fungal virulence and mycotoxins biosynthesis in *A. flavus*.

## 9. Conclusions

In the last decade, significant progress has been made in discovering the different molecules, factors, genes, and enzymes involved in each step of fungal development and aflatoxin biosynthesis in *A. flavus*. Interestingly, studies are constantly revealing new functions and molecular mechanisms of genes involved in the development of *A. flavus*, aflatoxin production, and pathogenesis. A full understanding of all these molecular mechanisms and their effects on the different genes encoding proteins in *A. flavus* is expected. Although there are many genes and molecular mechanisms regulating the fungal development and aflatoxins production in *A. flavus*, this study only considers the recently published studies (i.e., mainly in the past 10 years) focusing on the functions of the genes/proteins and molecular mechanisms contributing to *A. flavus* development and aflatoxins biosynthesis. In our future work, we intend to investigate the regulation mechanisms of different pathways on the morphogenesis and the secondary metabolites as well as the crosstalk between these signaling pathways. Additionally, advanced research on omics is highly warranted, as it could bring novel insights into fungal development and aflatoxins biosynthesis in *A. flavus*. For example, little information is currently known about the function of microRNAs in *A. flavus*. These small, non-coding RNA are particularly involved in gene silencing in numerous eukaryotic cells, which could play an essential function in the synthesis of fungal secondary metabolites in *A. flavus*. It is our hope that the information provided in this review may help our readers improve their understanding of gene function, signal transduction, genetic regulation, and other underlying molecular mechanisms in *A. flavus*, which may help in finding novel therapeutic strategies to control this common pathogen.

## Figures and Tables

**Figure 1 jof-07-00666-f001:**
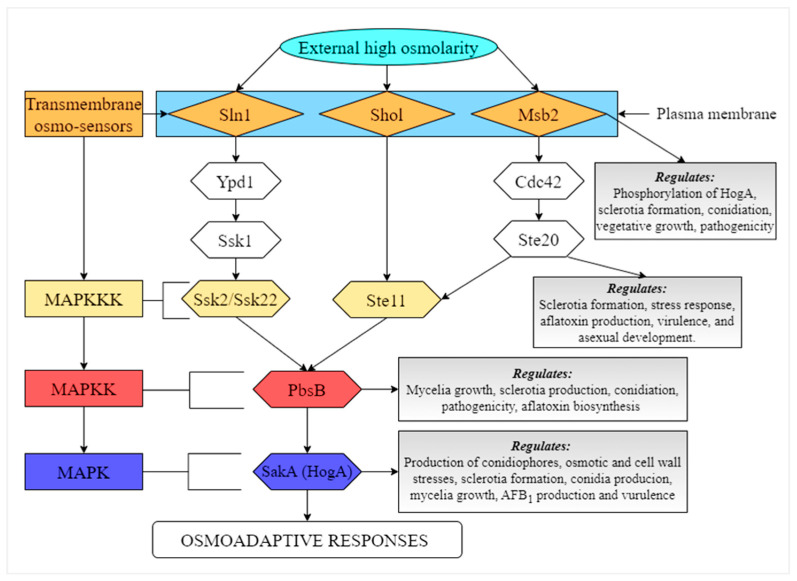
Illustration of the HOG-MAPK signaling pathway in *A. flavus.* PbsB integrates signals from two major independent upstream osmosensing mechanisms, which leads to the activation of specific MAPKKKs. Under osmostress, activated PbsB activates the SakA (HogA) MAPK, which induces a set of osmoadaptive responses.

**Figure 2 jof-07-00666-f002:**
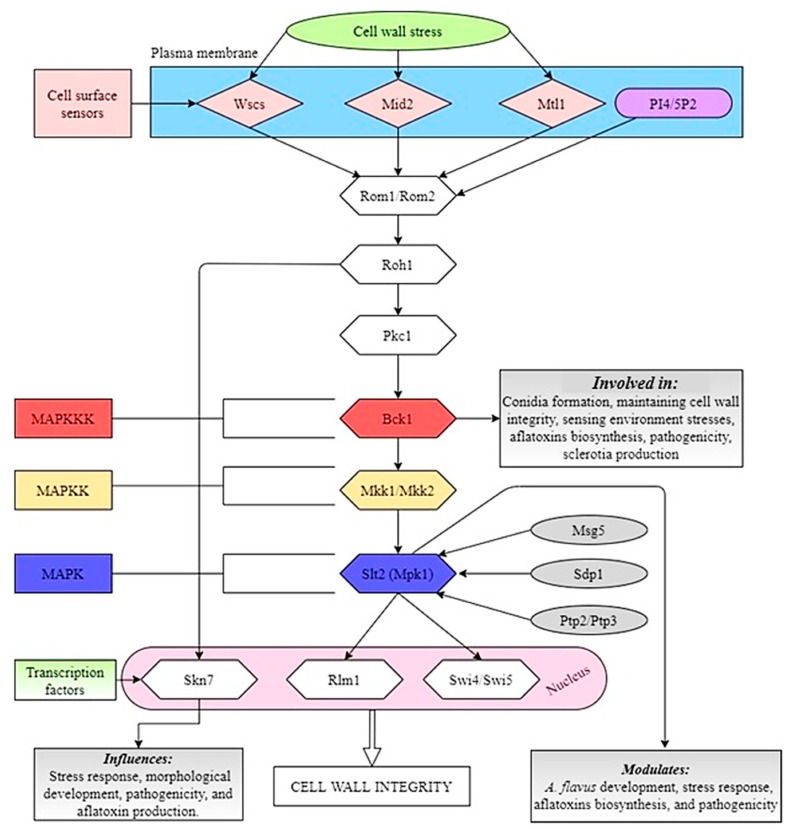
Illustration of the Slt2-MAPK signaling pathway in *A. flavus.* Signals are initiated at the plasma membrane (PM) through the cell-surface sensors Wscs, Mid2, and Mtl1. The extracellular domains of these proteins are highly O-mannosylated. Together with PI4, 5P2, which recruits the Rom1/Rom2 guanine nucleotide exchange factors (GEFs) to the plasma membrane, the sensors stimulate nucleotide exchange on Rho1. Two transcription factors, Rlm1 and SBF (Swi4/Swi6), are activated by the pathway. Skn7 may also contribute to the cell wall integrity transcriptional program.

## Data Availability

Not applicable.

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
