# Peer review of "Updates on the Functions and Molecular Mechanisms of the Genes Involved in Aspergillus flavus Development and Biosynthesis of Aflatoxins"

_jof, 2021, doi:10.3390/jof7080666_

Round 1

Reviewer 1 Report

This was a nice study to read, with a well-rounded literature review that comprised of 118 articles. Most of the articles cited are recent publications presenting state-of-the-art in the studied field. A few minor comments:

- line 100 and 101: "A research conducted by our research group on the genes functions indicated that several structure genes are also related to the aflatoxin 101production in A. flavus"  Does this refer to citation 26? It would be great to have the citation inserted in earlier stage

- line 465: "that involve in each step of the aflatoxin biosynthetic in 465A. flavus" - to be reviewed and rephrased

- line 476: "Although there are many genes and mechanisms in A. flavus" - mechanisms of what? I see that it is explained in the following sentence, but here it is too vague. The term "many genes" is also too vague. 

Reviewer 2 Report

jof-1234473-peer-review-v1

In this review, Tumukunde et al. reviewed recently published studies on function of genes and molecular mechanisms involving A. flavus development and secondary metabolites. In the abstract, the authors further define molecular mechanisms as posttranslational modifications, transcription factors, transcriptome, proteomics, metabolomics and transcriptome. The authors also state that the molecular mechanism of AF biosynthesis in A. flavus is still not well known (L54). Although I recognize the importance of this review, my overall feedback is that the review lacks focus and specifics – for example, how does understanding of gene function and molecular mechanisms can help in devising strategies to control AF production. For the reader, the authors should communicate clearly what is the key take home messages that this review wants to achieve. I have provided some feedback that may strengthen and refine the review.

Abstract.

This sentence lacks specifics “…the mechanism behind fungal development and AF biosynthesis in A. flavus is still unclear”

  • From my understanding, we know that fungal development is intimately linked (at the molecular level to AF biosynthesis, for example, transcription factors that regulate AF also regulate fungal development)
  • Authors may want to consider rephrasing this sentence and/or providing specifics to what is unclear in the mechanism?

Introduction.

  • L52-56; I am not sure if I agree with these statements
    • Authors might consider stating which fungal species (L53)
    • We know that although there exists evolutionary diversity of the toxigenic Aspergillus , the mechanism of AF synthesis is conserved (see for example, doi: 10.1002/ece3.3464)
  • I would recommend the authors provide a more specific list of goals for the review (L52-56 lacks specifics)
  • The introduction lacks the knowledge or consideration we have gathered in other closely related strains for example, paraciticus
  • It would also benefit the reader if the authors laid out a framework at the beginning of current research groups working on genomics and molecular mechanisms of AF biosynthesis in flavus.
  • ‘Recently discovered’ – how do authors define this – 5 years?

Section 2.

  • L110-111; repetition of word ‘together’

Section 4.

  • L160-161 …more accurately used to determine RNA expression – how ‘more accurately’?
  • L164-168 sentences are vague and lack specifics …it would help the reader if authors provided some specifics in the summary of how water activity and temperature affect AF biosynthesis
  • L171 – what is the function of 5-azacytidine?
  • This section could be improved by providing some conclusions on why and how this information is important to devising effective strategies to reduce and eliminate AF (authors vision for the review, L23-24)
  • L177 – what do authors mean by profound sequencing?
  • For example, ‘Yao et al….could authors summarize the major findings’?
  • Authors should consider rephrasing L180-186, why and how is this important?

Section 5.

-L190 – how does proteomics defeat the limitations of microarray – which limitation are authors referring to?

-L212-215 – ‘contribute to an improved understanding of this fungus’ – how does changes in gene product expression help devise strategies for reduction and elimination or control of development and sec. met?

-L233 – 5-azacytidine is mentioned again, authors assume that readers are familiar with this compound and its function

-L240-241 – ‘offer simultaneous and sensitive analysis’; I believe this phrase is vague – can authors provide more specifics – what does this mean and why is it more sensitive?

Section 7.

-L292-293 – similar comment as above “it is very important to characterize the role of enzyme regulating acetylation….”  Why is it important – how does this relate to AF biosynthesis and prevention strategies for reducing mycotoxins?

-L354-356 – how does this large dataset help researchers devise strategies to target and control aflatoxin production in A. flavus?

Conclusion.

-L473 -…availability of A. flavus genome sequence; I believe the sequence for NRRL3357 was published in 2015? Do authors mean metagenome?

-L472 is a little confusing – authors state that the genome sequence would enable advanced coverage of gene expression over different fungal growth (how?)

Other comments.

  • Figures – figure legends do not include any specifics explaining and outlining what the figures are communicating, lacks description. For example, in figure 1 – authors illustrate the HOG-MAPK signaling pathway in A. flavus but do not explain the colors and shapes used? Are these signaling molecules, transcription factors. Also, the goal of this review is focused on gene function and molecular mechanisms related to development and aflatoxin biosynthesis in A. flavus – however, figure 1 is lacking how these key players in signaling pathways are linked to development and secondary metabolite genes?
    • Also what do solid arrows represent; for example the arrow linking Sln1 to Ypd1 has a different function than the arrow that links SakA (HogA) to ‘Osmodaptive responses’
  • Similar comments for Figure 2

General notes.

  • Review seems very broad and lack specifics – multiple vague sentences
  • Bigger question is – how understanding molecular mechanisms can specifically develop effective strategies to limit A. flavus infection and AF production – we have known so much in the last 20-30 years – but AF contamination still occurs – is this strategy of understanding molecular mechanisms that regulate toxin production effective?
  • Authors did not consider the mol. mechanisms and gene functions in closely related toxigenic, aflatoxin producing species such as paraciticus?

Author Response

Response to Reviewer 2 Comments

Abstract.

Point 1: This sentence lacks specifics “…the mechanism behind fungal development and AF biosynthesis in A. flavus is still unclear”. From my understanding, we know that fungal development is intimately linked (at the molecular level to AF biosynthesis, for example, transcription factors that regulate AF also regulate fungal development). Authors may want to consider rephrasing this sentence and/or providing specifics to what is unclear in the mechanism?

Response 1: Thank you for your comment and explanation. We agree with you that this sentence lacks specifics. For clarification, the sentence has been re-phrased and the answer is on page 1 of the revised manuscript.

Introduction

Point 2: L52-56; I am not sure if I agree with these statements

Authors might consider stating which fungal species (L53). We know that although there exists evolutionary diversity of the toxigenic Aspergillus , the mechanism of AF synthesis is conserved (see for example, doi: 10.1002/ece3.3464)

Response 2: Thank you so much for this comment – This too has been addressed and the answer is on page 2 of the revised manuscript.

Point 3: I would recommend the authors provide a more specific list of goals for the review (L52-56 lacks specifics). The introduction lacks the knowledge or consideration we have gathered in other closely related strains for example, paraciticus. It would also benefit the reader if the authors laid out a framework at the beginning of current research groups working on genomics and molecular mechanisms of AF biosynthesis in flavus.

 ‘Recently discovered’ – how do authors define this – 5 years?

Response 3: Thank you for your comment. Introduction has been rewritten for clarification and better understanding.  The answer is on page 2 of the revised manuscript.

Section 2.

Point 4: L110-111; repetition of word ‘together’

Response 4: Thank you for your comment and we are sorry that this was repeated. The repeated word “together” has now been deleted and the answer to this comment is on page 3 of the revised manuscript.

Section 4

Point 5: L160-161 …more accurately used to determine RNA expression – how ‘more accurately’?

Response 5: Thank you for your comment. The sentences have been rephrased for clarification and better understanding. The answer is on page 4 of the revised manuscript.

Point 6: L164-168 sentences are vague and lack specifics …it would help the reader if authors provided some specifics in the summary of how water activity and temperature affect AF biosynthesis

Response 6: I agree with the reviewer that these sentences were not specific and could indeed confuse our readers. The sentences have now been re-phrased for clarification and the answer is on page 4 of the revised manuscript. 

Point 7: L171 – what is the function of 5-azacytidine?

Point 8: This section could be improved by providing some conclusions on why and how this information is important to devising effective strategies to reduce and eliminate AF (authors vision for the review, L23-24)

Response 8: Thank you so much for this comment. Indeed this information could have been added to the abstract but because the length of the abstract is limited in terms of word counts, we could not put all the information in the abstract. However, we decided that it would be better to answer and improve this on page 6 of the revised manuscript since the same question was also asked on point 12.  

Point 9: L177 –a) what do authors mean by profound sequencing?

Response 9: Thank you so much for your comment. This was actually our mistake and we apologize that we were not able to see it before sending in our first submission. The sentence has been corrected and the answer is on page 5 of the revised manuscript.

  1. b) For example, ‘Yao et al….could authors summarize the major findings’?

Response 9: Thank you so much for your comment. The sentence has been corrected and their major findings have been provided. The answer is on page 5 of the revised manuscript.

Point 10: Authors should consider rephrasing L180-186, why and how is this important?

Section 5

Point 11: L190 – how does proteomics defeat the limitations of microarray – which limitation are authors referring to?

Response 11: Thank you so much for your comment. We realised that the last part of the sentence was not accurate and we decided to remove it to keep the content simple and clear and the answer is on page 5 of the revised manuscript.

Point 12: L212-215 – ‘contribute to an improved understanding of this fungus’ – how does changes in gene product expression help devise strategies for reduction and elimination or control of development and sec. met?

Response 12: Thank you so much for this comment. The explanation has been improved and added to the manuscript for clarification. The answer is on page 6 of the revised manuscript.

Point 13: L233 – 5-azacytidine is mentioned again, authors assume that readers are familiar with this compound and its function

Point 14: L240-241 – ‘offer simultaneous and sensitive analysis’; I believe this phrase is vague – can authors provide more specifics – what does this mean and why is it more sensitive?

Section 7

Point 15: L292-293 – similar comment as above “it is very important to characterize the role of enzyme regulating acetylation….”  Why is it important – how does this relate to AF biosynthesis and prevention strategies for reducing mycotoxins?

Response 15: Thank you so much for this comment. The sentence has been changed and the answer is on page 7 of the revised manuscript.

Point 16: L354-356 – how does this large dataset help researchers devise strategies to target and control aflatoxin production in A. flavus?

Response 16: Thank you so much for this comment. This too has been answered and the answer is provided in a form of conclusion to all the 5 PTMs available on page 9 of the revised manuscript.

Conclusion

Point 17: L473 -…availability of A. flavus genome sequence; I believe the sequence for NRRL3357 was published in 2015? Do authors mean metagenome?

Response 17: Thank you so much for this comment. This too has been addressed and the answer is on page 12 of the revised manuscript.

Point 18: L472 is a little confusing – authors state that the genome sequence would enable advanced coverage of gene expression over different fungal growth (how?)

Response 18: Thank you so much for this comment. This was actually our mistake and we apologize that we did not fix it since our first submission. We have revised and improved this sentence (see page 12 of the revised manuscript).

Other comments.

Point 19: Figures – figure legends do not include any specifics explaining and outlining what the figures are communicating, lacks description. For example, in figure 1 – authors illustrate the HOG-MAPK signaling pathway in A. flavus but do not explain the colors and shapes used? Are these signaling molecules, transcription factors. Also, the goal of this review is focused on gene function and molecular mechanisms related to development and aflatoxin biosynthesis in A. flavus – however, figure 1 is lacking how these key players in signaling pathways are linked to development and secondary metabolite genes?

Also what do solid arrows represent; for example the arrow linking Sln1 to Ypd1 has a different function than the arrow that links SakA (HogA) to ‘Osmodaptive responses’

Similar comments for Figure 2

General notes.

Point 20: Review seems very broad and lack specifics – multiple vague sentences

Response 20: Thank you for your comment, the manuscript has been revised and all the necessary changes/revisions have been made accordingly.

Point 21: Bigger question is – how understanding molecular mechanisms can specifically develop effective strategies to limit A. flavus infection and AF production – we have known so much in the last 20-30 years – but AF contamination still occurs – is this strategy of understanding molecular mechanisms that regulate toxin production effective?

Response 21: Thank you so much for this comment. The answer for this question has been provided and the answer can be found on page 6 of the revised manuscript.

Point 22: Authors did not consider the mol. mechanisms and gene functions in closely related toxigenic, aflatoxin producing species such as paraciticus?

Response 22: Thank you so much for this comment. It is really a good question. However, since we only focused on Aspergillus flavus (Updated research on genes functions and molecular mechanisms for the development and biosynthesis of aflatoxins in A. flavus), adding information about other species would results in the paper losing its main focus making it difficult to draw a useful conclusion. We will probably look at that in our next study and see if we could conduct a comparative study focusing on these molecular mechanisms and gene functions in closely related toxigenic, aflatoxin producing species such as paraciticus.

Round 2

Reviewer 2 Report

The authors have provided a point-by-point response to my suggestions and comments. 

Author Response

Thank you so much for revising our manuscript. Language and spelling have been checked.